# Development and Testing of a New UWB Positioning Measurement Tool to Assist in Forest Surveys

**Ziyu Zhao** [1], **Zhongke Feng** [2,*], **Jiali Liu** [3] **and Yuan Wang** [2]

1    Department of Resource Management, Tangshan Normal University, Tangshan 063000, China
2    Precision Forestry Key Laboratory of Beijing, Beijing Forestry University, Beijing 100083, China
3    LuanNan County Natural Resources and Planning Bureau, Tangshan 063000, China
*    Correspondence: zhongkefeng@bjfu.edu.cn; Tel.: +86-138-1030-5579

**Abstract:** Forest resource inventory is a significant part of the sustainable management of forest ecosystems. Finding methods to accurately estimate the diameter at breast height (DBH), tree height and tree position is a significant part of forest resource inventory. The traditional methods of forest resource inventory are expensive, difficult, laborious and time-consuming; the existing systems are not convenient to carry, resulting in low working efficiency. In addition, it is usually necessary to rely on a forest compass, DBH taper and RTK or handheld GPS to set up the plot. These instruments each have a single function and cannot achieve accurate positioning under the forest canopy. Therefore, it is necessary to update the existing equipment and technology. This study aimed to design. a multi-functional, high-precision, real-time. positioning intelligent tree-measuring instrument that integrates plot the set-up, DBH measurement, tree height measurement and tree position measurement. The instrument is based on the ultra-wideband positioning principle, sensor technology, image processing technology, trigonometric functions, tree surveying and other related theories and realizes the functions of plot set-up, tree position measurement, DBH measurement, tree height measurement and other functions. The device was tested in four square plots. The results showed that the root mean squared. error (RMSE). of the tree position estimates ranged from 0.07 m to 0.16 m, while the relative root mean squared error (rRMSE) of the DBH estimates of individual trees ranged from 3.01 to 6.43%, which is acceptable for practical applications in traditional forest inventory. The rRMSE of the tree height estimates ranged from 3.47 to 5.21%. Furthermore, the cost of this instrument is only about one-third that of traditional forestry survey tools, while the work efficiency is three times that of the traditional measurement methods. Overall, the results confirmed that the tree measuring instrument is a practical tool for obtaining. accurate measurements of the tree position, DBH and tree height for forest inventories.

**Keywords:** UWB; forestry inventory; forestry plot; DBH; tree height

## 1. Introduction

Forest ecosystems are a significant part of terrestrial ecosystems and play a vital role in maintaining ecosystem balance of the entire biosphere [1]. They not only provide services such as soil and water conservation, climate regulation and biodiversity, but also provide wood, energy and food [2–4]. Due to the large area and complex, diverse structure of forests, it is generally not feasible to conduct a complete survey of the entire forest vegetation circle in actual survey work. Instead, the forest stand structure and changes are estimated by establishing forest plots and using sampling measurements [3]. Forest inventory mainly involves the clustering of forest resource information, and its purpose is to provide accurate estimates of the forest features, including wood volume, biomass or species richness in forest plots [5]. The distribution of tree locations in forest sample plots is a significant indicatrix for exploring the ecological forest structure and managing forest areas [4]. The

quick, convenient and accurate determination of tree coordinates in a sample plot survey is a significant component of forest inventory [5,6].

The efficiency and precision of forest inventory depend on the survey tools and methods. The traditional forest inventory usually utilizes a forest compass and tape measure to complete the sample land layout. Then, the forest resource inventory is completed via the tally principle. Traditionally, instruments such as a DBH caliper, hypsometer, TGC300 optical tree detector, etc., have been used to conduct forest resource inventory in China [7–9]. The published studies indicate that forest workers often record measurements manually [10,11], reducing the efficiency and increasing the possibility of measurement and data entry errors [12,13]. Although the accuracy of traditional measuring tools is relatively reliable, they are labor-intensive [14], while DBH and tree height measurements may lead to small errors for skilled workers. With the development of science and technology in recent years, including mathematical techniques of geographic information technology, computer technology and so on, multi-disciplinary research has led to the development of various new types of tree-measuring equipment and forestry observation technology. Moreover, new methods are constantly emerging, such as remote sensing, UAV photogrammetry, SLAM 3D laser scanning technology, etc. [15–27], which are already widely used in forest resource inventory work. For example, the total station was used to determine sample plots and measure the tree height. However, although the precision of the total station is relatively high, it cannot be popularized in the forest inventory due to its heavy weight and inconvenient to carry [28]. With the spread of GPS technology, workers also generally use some independent equipment to accomplish in measurement, such as mobile phones or handheld GPS devices to obtain geographic location data [23,24]. These devices are used to measure the four corners of the forest sample plot and the position of the tree stems. Other researchers used a combination of photogrammetry and GPS to visualize and calculate stand parameters [16]. Although GPS positioning methods can realize real-time positioning under the forest canopy, the propagation of the carrier phase signal is often subject to interference by the canopy, so the positioning accuracy is not satisfactory in forests with high canopy density. The first problem is that in addition to the need to hold a handset or GPS, the worker also needs to handle other instruments. Separately, resulting in a complicated process. In recent years, LiDAR technology has gradually become a fast and effective forest survey tool which is attracting worldwide attention [29]. TLS (terrestrial laser scanning) is generally the most mathematical remote sensing method to derive the specific forest inventory date at the stand plot level [19,20]. Other studies have explored the use of TLS at the plot level to assess the stem volume and biomass composition of individual tree reconstructions [30–32]. The disadvantages of this method are the high cost of data collection and processing [33], the need for professional software to process scan data and the need for technical staff, which is very difficult for grassroots forestry organizations to obtain. Other scholars have designed an arithmetic to measure the DBH and tree location using time-of-flight (TOF) cameras and camera pose point clouds through SLAM technology and the Lenovo mobile phone [25]. However, this approach can only be applied in favorable stand conditions since the survey accuracy may be affected if the bush ratio is too high.

In order to overcome the time.-consuming and laborious. nature of field surveys, it is urgent to develop portable, high-precision, low-cost and integrated forest inventory equipment. Sensor technology and computing technology make this possible. The ultra-wideband (UWB.) a ground-based technology that sends and receives. Carrier-less radio impulses using extremely accurate timing [34–39]. Compared with other wireless technologies, such as Wi-Fi, ZigBee and Bluetooth, it has high precision and is less affected by external occlusion [40]. Therefore, it is potentially suitable for relative positioning in the forest. This study mainly aimed to achieve the following objectives: (1) to develop a budget, polymorphous and real-time location intelligent tree-measuring instrument for a forest inventory; (2) to set up a plot and obtain the precise relative location of trees; and (3) to obtain the tree DBH and tree height.

The structure of the rest of this manuscript is as follows: In part 2, we provide the principle of the UWB positioning and technology of a UWB sample plot survey device, including the hardware design of the intelligent tree-measuring instrument and the workflow of the survey. In part 3, we provide a layout the sample plot and introduce the designed method, including a survey of the plot and the data acquisition method. In part 4, the instrument was tested in the study area, and the measuring results were verified and compared with the reference results. In part 5, the results of measuring results, working efficiency and cost of this manuscript are discussed. Lastly, part 6 concludes the paper.

## 2. Theory and Technology

### 2.1. UWB Positioning Technology

The UWB positioning theory is similar to the GPS positioning theory. As shown in Figure 1a, the positioning system consists of four anchors and one tag. Each anchor is equivalent to a base station in orbit, the tag is equivalent to a user side and the console is equivalent to a ground monitoring station. The anchor is used as a reference point and should be placed around the edge of the plot according to the size of the plot; the tag is used as a point to be positioned that has been integrated into the tree surveyor. By surveying the TOA (time of arrival.) from tag to anchor and multiplying it by the speed of light, the tag can acquire the distance to the anchor. Based on Figure 1b, we can develop four trilateration relations with each of the three circles (1-2-3, 1-2-4, 1-3-4, 2-3-4). Figure 1b is a schematic diagram of the trilateral positioning principle. Through the distances to four anchors and the coordinates of the reference anchors, multiple sets of spherical equations can be listed; then, the coordinates. Of the tag can be solved. Mathematically. Assuming that the coordinates of the tag are $(X_n, Y_n)$, the coordinates of the anchor are $(x_i, y_i)$ and $d_i$ is the distance between the anchor and the tag, according to the TOA ranging algorithm, the coordinates of $p_1(X_1, Y_1)$ can be obtained as follows:

$$\begin{cases} 2(x_1 - x_2)X + 2(y_1 - y_2)Y = d_2^2 - d_1^2 + x_1^2 - x_2^2 + y_1^2 - y_2^2 \\ 2(x_2 - x_3)X + 2(y_2 - y_3)Y = d_3^2 - d_2^2 + x_2^2 - x_3^2 + y_2^2 - y_3^2 \\ 2(x_1 - x_3)X + 2(y_1 - y_3)Y = d_1^2 - d_3^2 + x_1^2 - x_3^2 + y_1^2 - y_3^2 \end{cases} \tag{1}$$

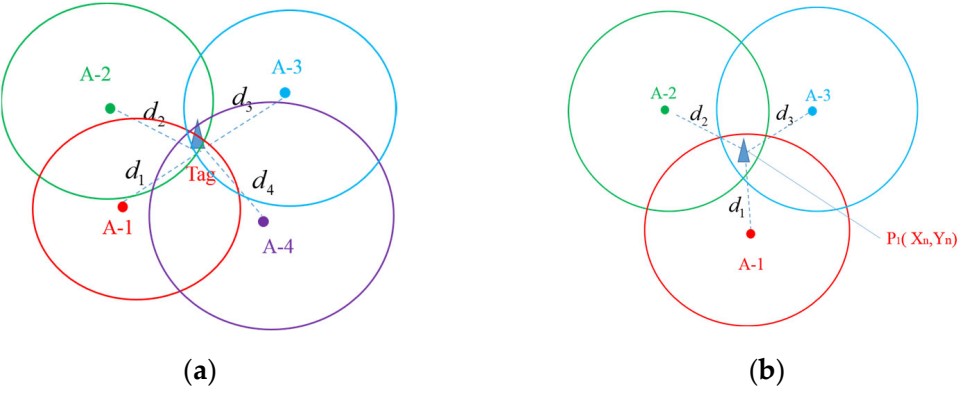

|  (a)  |  (b)  |

**Figure 1.** Schematic representation of the trilateral UWB location. (**a**) illustrates the position estimation, (**b**) illustrates the trilateration location theory.

According to Equation (1), the additional values of $p_2(X_1, Y_1)$, $p_3(X_1, Y_1)$ and $p_4(X_1, Y_1)$ can be calculated. In this paper, the coordinates of the four base station points were adopted,

and the RSSI theory was used to calculate the location of the tag [40]. The coordinates $(X_n, Y_n)$ can be calculated using the Equation (2).

$$
\begin{cases}
X_n = \dfrac{\frac{x_1}{d_1^2+d_2^2+d_3^2} + \frac{x_2}{d_1^2+d_2^2+d_4^2} + \frac{x_3}{d_1^2+d_3^2+d_4^2} + \frac{x_4}{d_2^2+d_3^2+d_4^2}}{\frac{1}{d_1^2+d_2^2+d_3^2} + \frac{1}{d_1^2+d_2^2+d_4^2} + \frac{1}{d_1^2+d_3^2+d_4^2} + \frac{1}{d_2^2+d_3^2+d_4^2}} \\[4mm]
Y_n = \dfrac{\frac{y_1}{d_1^2+d_2^2+d_3^2} + \frac{y_2}{d_1^2+d_2^2+d_4^2} + \frac{y_3}{d_1^2+d_3^2+d_4^2} + \frac{y_4}{d_2^2+d_3^2+d_4^2}}{\frac{1}{d_1^2+d_2^2+d_3^2} + \frac{1}{d_1^2+d_2^2+d_4^2} + \frac{1}{d_1^2+d_3^2+d_4^2} + \frac{1}{d_2^2+d_3^2+d_4^2}}
\end{cases}
\tag{2}
$$

### 2.2. The Technology of a UWB Sample Plot Survey Device

#### 2.2.1. Hardware Design of Intelligent Tree-Measuring Instrument

Aiming to overcome the problems of high cost, low measurement efficiency, inconvenient data transmission, and the complicated operation of existing measurement equipment, a smart tree-measuring instrument based on UWB positioning (hereafter referred to as tree-measuring instrument) has been designed, as shown in Figure 2a. The total weight of the instrument is not more than 2 kg. The hardware components of the tree-measuring instrument include a Personal Digital Assistant (PDA), laser ranging module, UWB positioning device, gyro altitude sensor and battery. When the tree-measuring instrument is working, four attached positioning devices are needed. The additional anchors UWB positioning devices are shown in Figure 2b. The detailed parameters and hardware structure design are shown in Appendix B. The PDA acts as the data processing, transmission and human–computer interaction module of the tree-measuring instrument, providing an operating software environment and controlling the interaction with other hardware modules. The UWB positioning device is a real-time centimeter-level positioning module, which can realize the real-time positioning and acquisition of relative coordinates under the forest canopy. The three-axis gyroscope and three-axis accelerometer are inclination measurement modules. The laser rangefinder is used to measure the distance between the instrument and the stem when performing the DBH function in the forest, and the distance between the dendrometer location and the stem is used to calculate the DBH in the photo. The photogrammetric. Platform is used to place and fix the tree detector, and the platform can rotate 360 degrees when working.

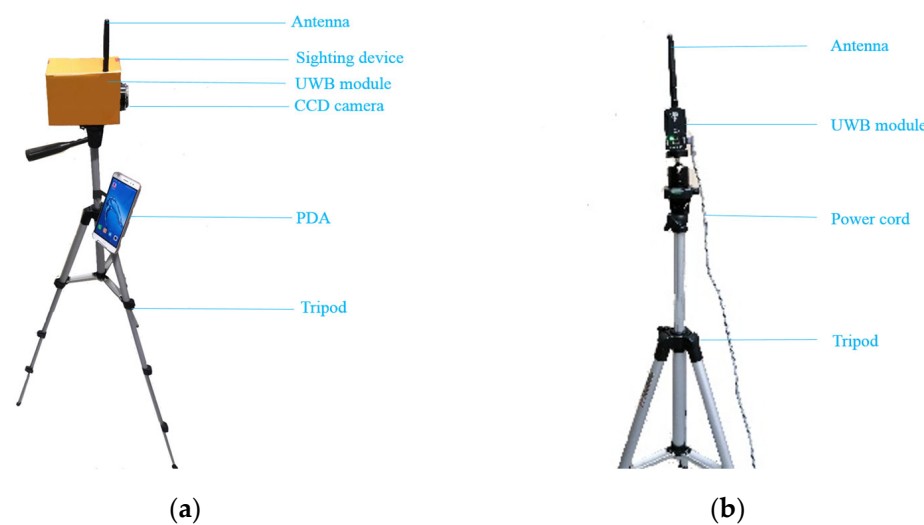

**(a)**　　　　　　　　　**(b)**

**Figure 2.** Overall design of the tree-measuring instrument and additional anchors. (**a**) Diagram of tree-measuring instrument, (**b**) Diagram of additional anchors.

#### 2.2.2. Workflow

The software part mainly includes PDA development and design. It adopts modular structure processing, and its embedded software was compiled in the Android Studio3.0

development environment based on the Java language, mainly including the plot layout module, DBH measurement module, tree height measurement module, stand measurement module and data calculation module. The overall control of the program is shown in Figure 3. According to the situation on the scene, the user selects different measurement programs and connects the measurement module through Bluetooth and Wi-Fi.

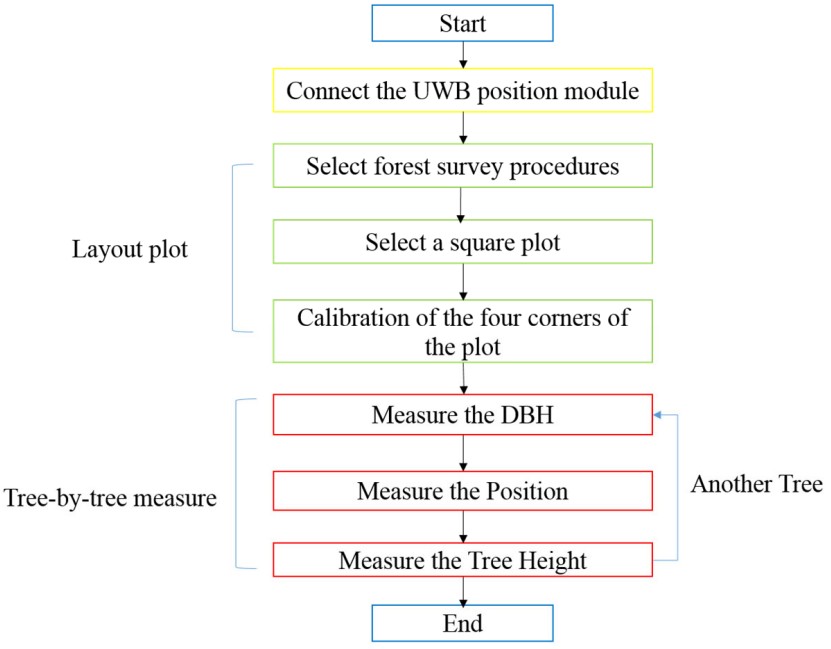

**Figure 3.** The workflow of the tree-measuring instrument for data acquisition.

The measurement results are stored in an SQLite database to enable the data management and data analysis functions. The data management functions include the data recording function: recording the position information, DBH, tree height, tree species and other information of the measured tree; data query function: the operator can find similar data (such as the location, DBH, tree height value range, etc.); data editing function: the operator can edit the measured data (including adding, deleting, revising, etc.); data can be saved into Excel through the data export function.

Specifically, before working with the tree-measuring instrument, we first assembled the equipment. Herein, modular instrument assembly saves time and labor, and in our field measurements, we only needed to install the tree-measuring instrument and additional anchors on the tripod and fix it with screws. Then, we needed to attach four anchors placed around the sample area, and we turned on the power of each anchor and tree-measuring instrument. The connection between each module is automatic, and we did not need to set other parameters when working in the field because we had set the parameters during the assembly. We therefore only needed to set the parameters the first time we used the instrument. Secondly, the system supports one-key calibration of all anchors (note: the four anchors need to be placed at the same height, otherwise there will be additional calibration errors). Then, through the console interface of the "coordinate measuring" button, we can see the coordinates of the anchor in the console and screen icon gradually converge. When the coordinate calibration is successful, the system automatically exits the "one-key" calibration mode. The time required for the calibration of anchors is less than one minute. At this point, the system settings have been completed, and the next tree measurement can be carried out. The calibration theory of the anchors is illustrated in Figure 4 and Equation (3). The point O is the location of anchor $A_1$, and the x-axis is the projection direction from anchor $A_1$ to $A_4$. The coordinates of anchors $A_1$, $A_2$, $A_3$ and $A_4$ can then be obtained as shown in Equation (3).

$$(x_1, y_1) = (0, 0)$$
$$(x_2, y_2) = \left( \frac{d_{14}^2 + d_{12}^2 - d_{24}^2}{2d_{14}}, \sqrt{d_{12}^2 - \left( \frac{d_{14}^2 + d_{12}^2 - d_{24}^2}{2d_{14}} \right)^2} \right)$$
$$(x_3, y_3) = \left( \frac{d_{14}^2 + d_{13}^2 - d_{43}^2}{2d_{14}}, \sqrt{d_{13}^2 - \left( \frac{d_{14}^2 + d_{13}^2 - d_{43}^2}{2d_{14}} \right)^2} \right)$$
$$(x_4, y_4) = (d_{14}, 0)$$

(3)

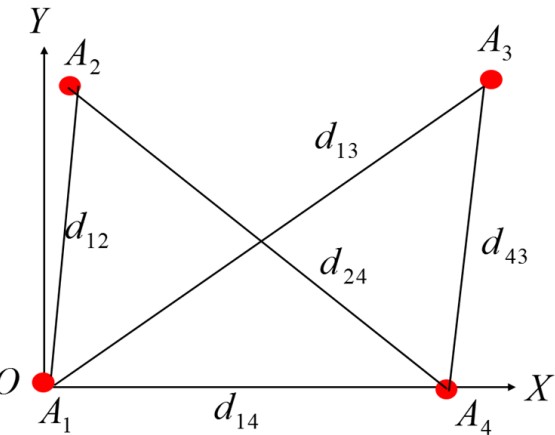

**Figure 4.** Schematic diagram of the anchors calibration principle, $A_1 \sim A_4$ are the anchors locations; $d_{ij}$ are the distance between anchors $A_1 \sim A_4$.

## 3. Materials and Methods

### 3.1. Study Area

The experiment was performed in Xishan Park in the mountainous area west of Beijing city, China (39° 58′ 12″ N, 116° 12′ 10″ E; Figure 5). This forest is man-made and was planted with *Platycladus orientalis*, *Cotinus coggygria* Scop, *Pinus tabuliformis*, *Quercus variabilis* Blume, *Ginkgo biloba* and *Fraxinus chinensis* Roxb as the major species. We sampled four 25.82 × 25.82 m square (677 m²) plots. There were only a few bushes and herbs in the forest. The basic descriptions of the plots are shown in Table 1.

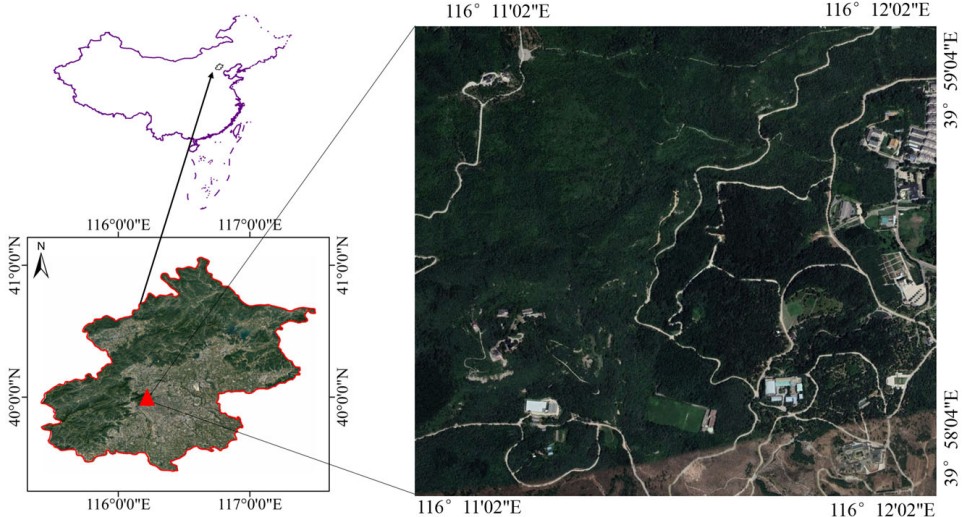

**Figure 5.** Location of the study area.

**Table 1.** Statistical summary of the trees in the study plot.

| Plot | Tree Number | Species | DBH (cm) | | | Tree Height (m) | | |
|---|---|---|---|---|---|---|---|---|
| | | | **Mean** | **Min** | **Max** | **Mean** | **Min** | **Max** |
| 1 | 40 | *Pinus tabuliformis* | 19.2 | 14.3 | 27.5 | 8.8 | 5.8 | 13.8 |
| 2 | 44 | *Platycladus orientalis* | 19.1 | 7.8 | 37.4 | 9.5 | 5.8 | 14.2 |
| 3 | 52 | *Quercus variabilis* | 16.9 | 7.6 | 36.5 | 9.1 | 6.2 | 14.2 |
| 4 | 42 | *Ginkgo biloba* | 29.1 | 20.6 | 37.3 | 11.8 | 8.7 | 16.4 |

*3.2. Methods*

3.2.1. Layout the Sample Plot

Before laying the sample plot, we first set up a communication connection between the tag and the anchors. The modules were connected automatically, and the positioning network was established. Subsequently, four anchors were arranged according to the size of the sample plot area. If the plot size is large (667–900 m$^2$), the distance between the anchors must also increase, and vice versa [41]. After placing the anchors, the precise measurement of the coordinates of the 4 base stations can be automatically realized using the console. After the automatic measurement of the coordinates of the anchors was completed, the tree-measuring instrument can realize the accurate relative positioning under the forest canopy. In order to facilitate the operation, a relative plane rectangular coordinates system was set in the system, as shown in Figure 6a. The X-axis is the line between A1 and A4, and the default A1 coordinate of the system is (x = 0, y = 0). The x value increases from point A1 to point A4. The y-axis is a line perpendicular to the x-axis passing through A1.

The specific process of plot layout was as follows:

(1) We selected the size of the plot area (such as 100, 400, 667, 900 m$^2$ or custom) to measure using the PDA. The shape of the plot was usually square. After selecting the size of the plot, we determined the position of the first corner point of the plot.

(2) After the first corner point was laid, the other three corner points were virtualized on the screen of the PDA, after which the system prompted the direction and distance to move to the second corner point until the final corner point was laid.

(3) We repeated the steps 1 and 2 until all corners of the sample plot were laid out.

(4) After returning to the position of the first corner point, we measured the position of the first corner point again; then, the system calculated the position error of corner points of the first plot in two measurements automatically. If the perimeter of plot closure difference was less than 0.5%, then the forest investigation was carried out (as shown in Figure 6b).

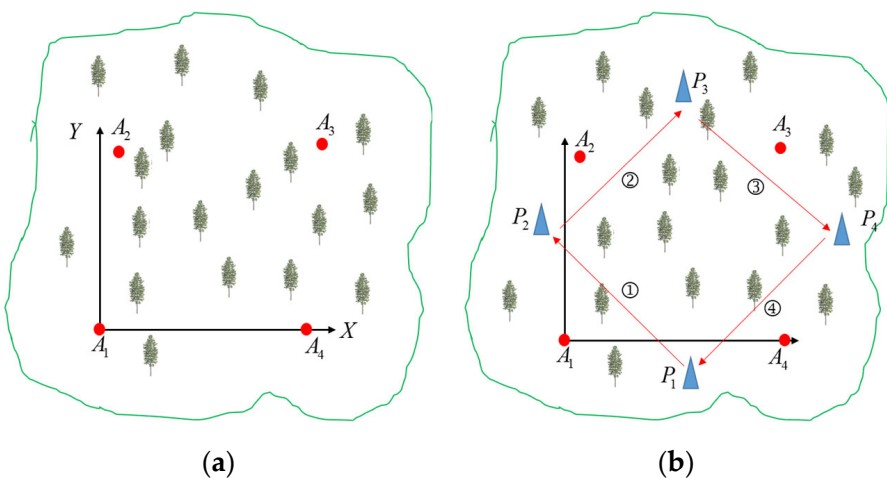

(**a**)  (**b**)

**Figure 6.** Schematic diagram of laying sample plots. (**a**) Schematic diagram of the plane coordinate system; (**b**) process of setting up a sample plot. $A_1 \sim A_4$ are the anchors location; $P_1 \sim P_4$ are the corner points of the forest sample plot; ①~④ are the process steps for setting up a sample plot.

### 3.2.2. Measurement of the DBH

Before carrying out the sample plot investigation, in order to accurately settle the coordinate information of the polygon point of the photographic image and achieve accurate mapping, the coordinates of the measuring center point must be accurately matched with the high-resolution image point by the CCD lens under the control of the tree-measuring instrument. This requires a strict and comprehensive inspection of the assembled instrument before using the instrument to determine the altitude offset parameters of the center of the CCD lens relative to the center of the tree-measuring instrument. The relative coordinates $(T - XYZ)$ of the center of the tree-measuring instrument are used as the standard coordinates, which are necessary to determine the displacement $(e_x, e_y, e_z)$ and rotation angle $(\varepsilon_x, \varepsilon_y, \varepsilon_z)$ of the center of the CCD lens relative to the center coordinates of the tree-measuring instrument.

After determining the coordinate position data $(X_i, Y_i, Z_i)$ and photogrammetry data $(X_i', Y_i', Z_i')$ of the tree-measuring instrument at the same time, the corresponding coordinates have a certain relationship of rotation and translation, as shown in Equation (4).

$$\begin{bmatrix} X_i \\ Y_i \\ Z_i \end{bmatrix} = \begin{bmatrix} X_i' \\ Y_i' \\ Z_i' \end{bmatrix} + \begin{bmatrix} 0 & -Z_i' & Y_i' \\ Z_i' & 0 & -X_i' \\ -Y_i' & X_i' & 0 \end{bmatrix} \begin{bmatrix} \varepsilon_x \\ \varepsilon_y \\ \varepsilon_z \end{bmatrix} + \begin{bmatrix} e_x \\ e_y \\ e_z \end{bmatrix} \tag{4}$$

The coordinate displacement $(e_x, e_y, e_z)$ and rotation angle $(\varepsilon_x, \varepsilon_y, \varepsilon_z)$ of the CCD lens center corresponding to the center of the tree-measuring instrument were determined using Equation (6) and corresponding coordinate position data, and the relative position relationship between the two measurement centers was obtained. Then, the relative offset transformation relationship between the tree-measuring instrument coordinate system and the photogrammetry data was established. The comprehensive calibration of the instrument only needs to be tested once before starting the forest inventory. The positional relationship between the center of the CCD lens and the tree-measuring instrument's measurement center was unchanged. It was not necessary to check it again before further use.

For the DBH measurement, the distance ($L$) from the tree-measuring instrument to the stem was measured by a laser range finder, the fixed focal length of the CCD lens of the instrument was $f$, and the DBH was calculated according to the theory of photogrammetry, as shown in Equation (5).

$$DBH = \frac{NL}{f} \tag{5}$$

Note: $N$ is the number of pixels, $L$ is the length from the tree-measuring instrument to the stem and $f$ is the fixed focal length of the CCD lens.

The specific steps of DBH measurement are as follows:

(1) Adjust the instrument until the reticule in the screen aims at the center of the stem; click on distance measurement; click on the left upper corner of the screen distance $L$ update "0.000"; then click the red button to confirm, and an interface sketch of the DBH measurement as shown in Figure 7a will appear.

(2) Adjust the "+" and "−" buttons until the border line is placed on both sides of the stem, and then the system automatically calculates the DBH. The measured results will be automatically displayed and stored. Click "Results" to view the measurement results; the DBH results interface as shown in Figure 7b will appear.

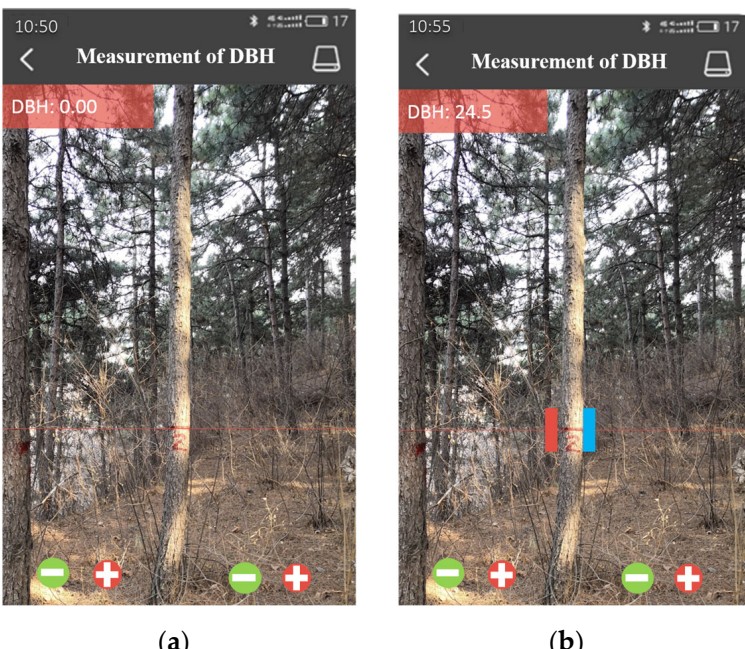

<div align="center">(<b>a</b>)                                (<b>b</b>)</div>

**Figure 7.** Schematic diagram of setting up a sample plot. (**a**) Interface of the DBH measurement; (**b**) interface of the DBH measurement result. The red and blue square are used to mark the DBH.

### 3.2.3. Measurement of the Position

After the measurement of the DBH, the position of the tree-measuring instrument did not need to be changed. We clicked the coordinate measurement button on the screen to collect the position of the measurement site $(x_c, y_c)$. At this time, the measurement site position was obtained. Then, we adjusted the tree finder until the laser rangefinder was aimed at the stem and clicked the "range" button on the screen. The instrument recorded the distance $(S_c)$ from the measurement position to the stem and the azimuth $(\beta_c)$. According to the principle of coordinate solution in surveying and mapping, the position $(x_i, y_i)$ of the tree stem was calculated.

### 3.2.4. Measurement of the Tree Height

(1) After measuring the position of the tree trunk, the relative coordinates of the tree stem were recorded as $(x_i, y_i)$ (only the plane coordinates were measured during the survey, and the coordinates of height Z were not considered, so Z = 0). At this time, the tree-measuring instrument was moved to a place a little further away (3~10 m) from the measured tree, as shown in Figure 8.

(2) After setting up the instrument, we clicked the measurement position button on the screen, and the distance $(L)$ from the instrument to the tree stem was calculated automatically using the tree stem position and the instrument position. The calculation principle is shown in Equation (6).

$$L = \sqrt{(x'_i - x_i)^2 + (y'_i - y_i)^2} \tag{6}$$

Note: $x_i, y_i$ is the position of the stem; $x'_i, y'_i$ is the position of the tree-measuring instrument.

(3) Finally, we used the instrument to aim at the root of the tree, clicked "ok" on the screen to measure the downward inclination $\alpha_1$; then, we turned the instrument up, aimed at the treetop and clicked "finish" on the screen. The tree height $L$ of the measured tree was automatically displayed at this time. The calculation principle is shown in Equation (7).

$$H = L \cdot (\tan \alpha_1 + \tan \alpha_2) \tag{7}$$

$\alpha_1, \alpha_2 \in (-90° \sim 90°);$

Note: $L$ is the distance of the instrument position to the stem position;
$\alpha_1$ is the downward inclination (°);
$\alpha_2$ is the upward inclination (°).

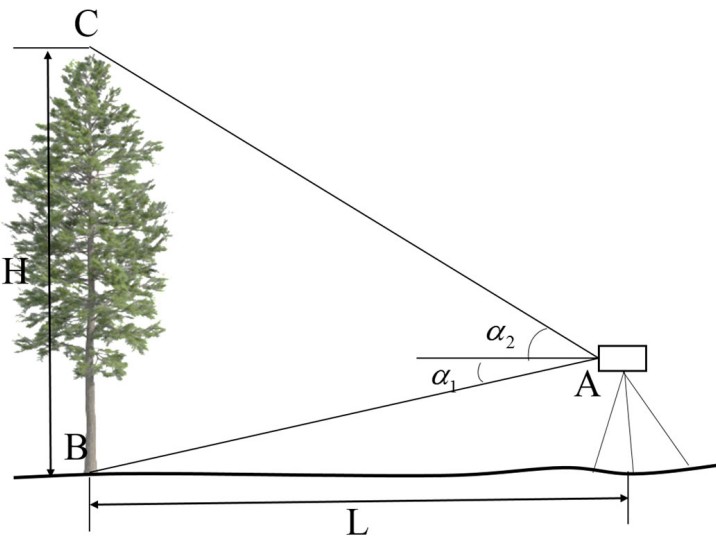

**Figure 8.** Schematic representation illustrating the principle of tree-height measurement.

3.2.5. Evaluation of the Accuracy of the Stem Position, DBH and Tree Height Measurements

The DBH, height and tree position in the forest plot were measured as reference values. A DBH caliper and an NTS-391 total station (Southsurvey Guangzhou, China) were used to measure the DBH, tree position and tree height in the forest stand sample plots; the tree height was measured by the suspension mode of the total station. The measure of positional accuracy is the percentage of trees detected relative to their position [36–38]. The bias, root mean squared measurement error (RMSE), relative bias (rBias) and relative RMSE (rRMSE), which were calculated by Equations (8)–(11), respectively, were employed to verify the precision.

$$Bias = \frac{1}{n}\sum_{i=1}^{n} e_i = \frac{1}{n}\sum_{i=1}^{n}(y_i - y_{ri}) \tag{8}$$

$$RMSE = \sqrt{\frac{\sum(y_i - y_{ri})^2}{n}} \tag{9}$$

$$rBias = \frac{Bias}{\overline{y_r}} \times 100\% \tag{10}$$

$$rRMSE = \frac{RMSE}{\overline{y_r}} \times 100\% \tag{11}$$

where $y_i$ is the $i$-th measure values, $y_{ri}$ is the $i$-th reference, $\overline{y_r}$ is the mean of the reference values and $n$ is the number of measure values. The results of the mapping were evaluated with relative bias and RMSE and calculated by dividing the mean of the reference value by the bias and RMSE.

## 4. Results

### 4.1. Evaluation of Tree Position

In order to test the precision of the tree-measuring instrument under the forest canopy, the position of the tree determined by the NTS-391 total station was used as the reference value. After obtaining the high-precision tree position with the tree-measuring instrument, it was compared with those measured using the total station. The estimated positions of four plots are shown in Table 2 and Figure 9. After extracting the tree positions, the

maximum bias in the four test plots was 0.02 m, the minimum bias was 0.01 m, the highest RMSE was 0.16 m and the lowest RMSE was 0.07 m. The RMSE values in the x-axis were 0.07~0.16 m, and RSME values in the y-axis were 0.09~0.12 m. As shown in Figure 10, the positioning system had a weak error in four plots, but the overall standard deviation in the maximum variability direction was less than 0.15 m, which is an acceptable systematic error.

**Table 2.** Comparison of the tree positions obtained using the tree-measurement instrument estimates and field measurements.

| Plot Number | Bias (m) | | RMSE (m) | |
|---|---|---|---|---|
| | X | Y | X | Y |
| 1 | 0.02 | −0.01 | 0.16 | 0.12 |
| 2 | 0.01 | 0.01 | 0.11 | 0.09 |
| 3 | 0.02 | 0.02 | 0.14 | 0.09 |
| 4 | 0.02 | 0.01 | 0.07 | 0.11 |

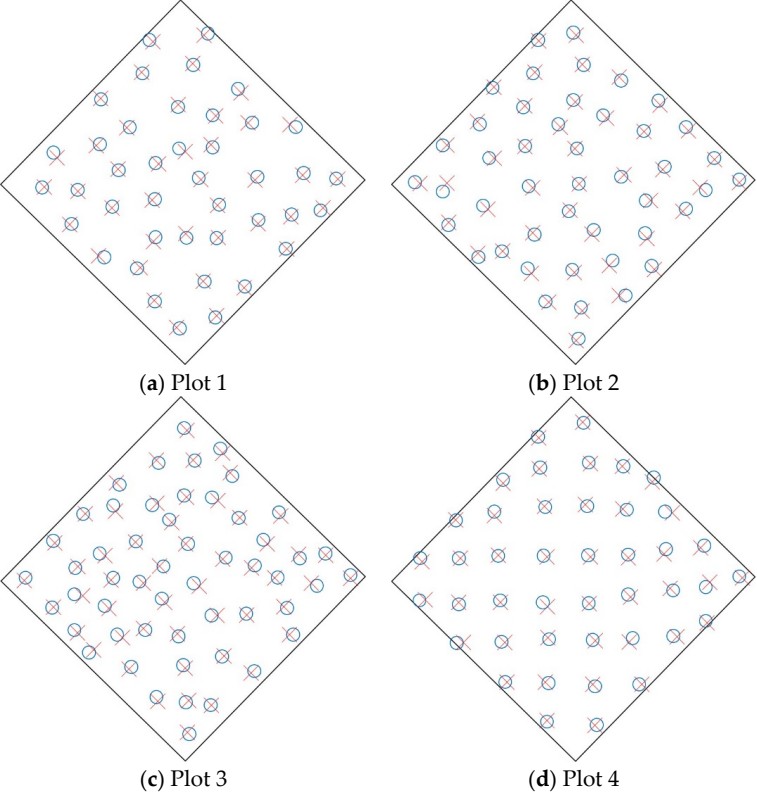

(**a**) Plot 1　　　　　　　　　(**b**) Plot 2

(**c**) Plot 3　　　　　　　　　(**d**) Plot 4

**Figure 9.** Estimated and reference stem positions: the blue circles indicate the stem references; the red crosses indicate the estimates.

### 4.2. Evaluation of DBH

A DBH caliper was used to survey the DBH in the four plots, and these manual measurement results were used as the reference values. At the same time, the tree-measuring instrument was used to measure the DBH of the trees in the four plots. The results are shown in Table 3. The DBH precision for the four plots was acceptable since the maximum value of rBias was 2.05% and the minimum value was −0.38%; the maximum value of rRMSE was 6.43%, and the minimum. value was 3.01%. This precision met the requirements of forest inventories [41].

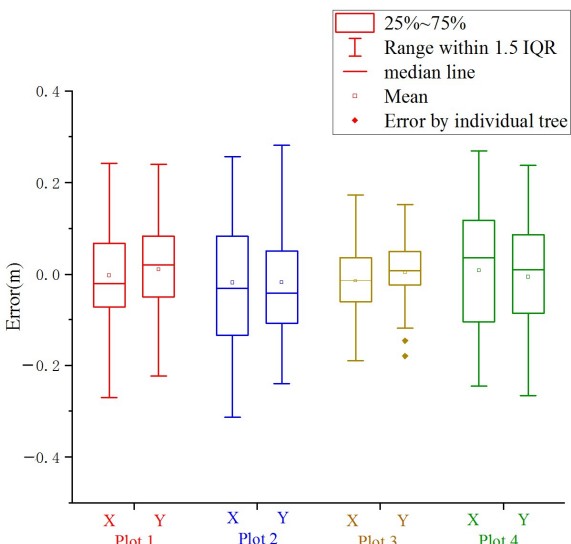

**Figure 10.** The position errors of all trees in the plots.

**Table 3.** Comparison of the DBH estimates produced using the tree-measuring instrument and manual field measurements.

| Sample Number | Bias (cm) | rBias (%) | RMSE (cm) | rRMSE (%) |
|---|---|---|---|---|
| 1 | −0.14 | −0.71 | 1.23 | 6.43 |
| 2 | −0.07 | −0.38 | 0.78 | 4.09 |
| 3 | 0.38 | 2.05 | 1.01 | 5.99 |
| 4 | 0.18 | 0.61 | 0.88 | 3.01 |

Figure 11 shows the precision of the tree-measuring instrument for DBH in the four plots, with the horizontal axis representing the reference value and the vertical axis representing the estimated value. The $R^2$ of the four plots was distributed at 0.978, indicating that the linear fitting precision of the estimation and reference values was good. As shown in Figure 11, the estimates were distributed on either side of the reference value, and the average RMSE values for DBH in the four plots was 0.948 cm.

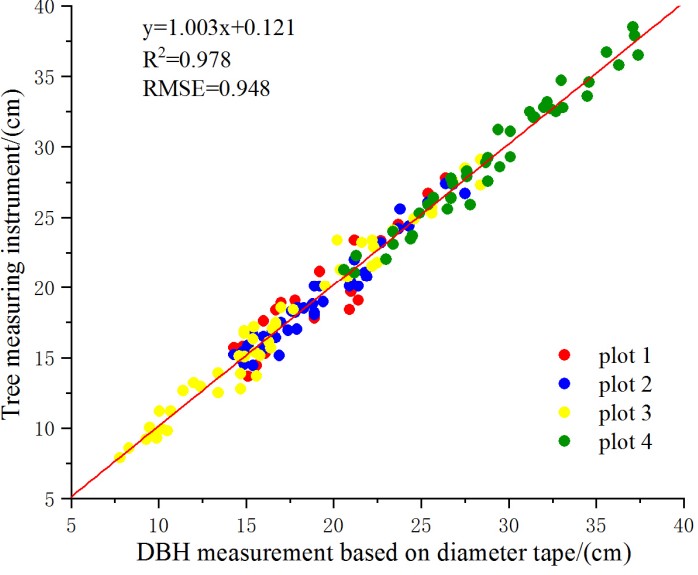

**Figure 11.** Comparative analysis of DBH estimation accuracy in the four plots.

### 4.3. Evaluation of Tree Height

A total station instrument was used to survey the tree height in the four plots, and the measurement results were used as the reference values. As shown in Table 4, the tree height precision of the four plots was acceptable, with a maximum rBias value of 2.36% and a minimum value of 0.46%; the maximum value of rRMSE was 5.21%, and the minimum value was 3.47%.

**Table 4.** Comparison of the tree height estimates produced using the tree-measuring instrument and total station.

| Plot Number | Bias (m) | rBias (%) | RMSE (m) | rRMSE (%) |
|---|---|---|---|---|
| 1 | 0.05 | 0.51 | 0.45 | 5.21 |
| 2 | −0.09 | −1.02 | 0.33 | 3.47 |
| 3 | 0.22 | 2.36 | 0.32 | 3.56 |
| 4 | 0.06 | 0.46 | 0.58 | 4.94 |

In order to verify the accuracy of the tree-measuring instrument for tree height, the reference values of the four plots were arranged from small to large. Figure 12 shows the accuracy of the tree-measuring instrument in the four plots, with the horizontal axis describing the reference value and the vertical axis describing the estimate value. The $R^2$ values of the four plots were distributed at 0.969, and the RMSE values of tree height was 0.46 m, indicating that the linear fitting precision of the estimation and reference value was good.

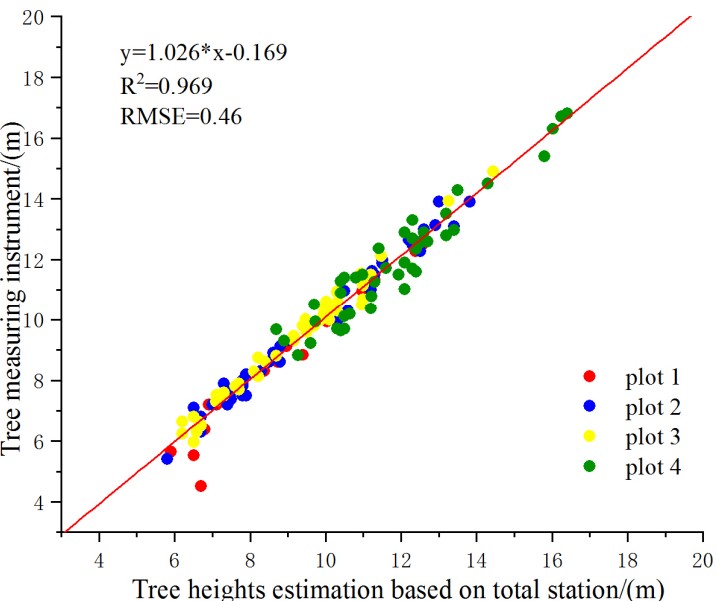

**Figure 12.** Comparative analysis of tree height measurement in the plots.

### 4.4. Comparing the Efficiency of Different Methods

In Table 5, the measuring instrument presented in this research was compared with the current survey tools. First, in terms of time, the method developed in this paper can quickly and accurately measure tree position, DBH and tree height simultaneously via non-contact measurement. There were four plots in this experiment, and the time consumed to measure each plot was recorded. The results show that the time required to complete each plot was two- or three-times shorter than that of traditional measurement methods. Second, in terms of cost, the tree-measuring instrument presented in this paper has the characteristics of high integration and portability, while the cost of this instrument was only about one-third that of traditional forestry surveys tools. It only requires a set of equipment

to complete data collection and storage, eliminating the need for complicated paper quality records. Finally, when comparing the number of people required by the two methods, the instrument only requires one person to complete the forestry survey independently, while the traditional method requires at least three people to assist in the forestry surveys, including the assembly of the instrument, set up the plot, the measurement of the position, the measurement of the DBH and the measurement of the height of the tree. The price of each accessory of the tree-measuring instrument is listed in Appendix B.

**Table 5.** Comparison of the efficiency and cost between this instrument and traditional instruments.

| Work Mode | Tree Position | DBH | Tree Height | Cost | Number of Investigators | Measurement Time |
|---|---|---|---|---|---|---|
| Traditional method | Forest compass (300 dollars) or RTK (3000 dollars) | DBH caliper (10 dollars) | Hypsometer (700 dollars) or total station (1200 dollars) | 1000~4200 dollars | ≥4 | 120~150 min |
| Presented method | Tree-measuring instrument | | | 600 dollars | 1 | 45~60 min |

## 5. Discussion

UWB technology provides a solution that does not require GNSS signal positioning in forests and other places. In this study, we used UWB positioning technology similar to the GPS anchor positioning mode to provide accurate positioning under the forest canopy. On the other hand, UWB technology does not use a carrier wave, but transmits data by sending nanosecond non-sinusoidal narrow pulse signals. At the same time, the acceptor of the UWB system is unlike the traditional acceptor since it does not need medium-frequency processing. Therefore, the structure of the UWB system is relatively simple [35]. UWB technology has extremely strong penetrating ability, enabling it to perform precise positioning indoors and underground [36–38], while GPS can only work within the coverage of GPS positioning satellites [23,24]. Unlike the absolute geographic position of GPS, UWB radio locators can provide relative positions with positioning accuracy up to the centimeter level. In this study, we took advantage of the UWB positioning principle to design and develop a set of low-cost forest survey equipment for the needs of forestry surveys. At the same time, the tree position, DBH and tree height were measured in real time using the laser ranging principle, photogrammetric principle and coordinate azimuth calculation principle, respectively. The position of the tree stem was calculated from the azimuth and distance of the instrument to the tree stem, which avoids signal blockage caused by the direct contact between the instrument and the tree. In the forest survey, a square standard plot was first established using relative UWB technology, and the four corner offset errors of the plot were detected by loop closure. The results indicate that the biases of the X and y axes were between −0.01 and 0.02 m, while the RMSE values of the x and y axes were from 0.07 to 0.16 m and from 0.09 to 0.12 m, respectively. The main reason for the location errors is a systematic error in the location system. The transmission signal may be affected by epiphytes, or thick understory, which may occlude the tree crowns to a certain degree. The combination of various factors led to the location error. Previous studies [21,42,43] have only relied on mobile phone positioning, and the location RMSE was between 5 and 8 m. Compared with traditional methods [16,23,24], which rely on devices such as traditional GPS or RTK signals that can be easily blocked by the tree canopy, this method has the advantages of high integration and high precision.

On the basis of ground photogrammetry, the DBH of trees was quickly estimated. The results show that the rRMSE values of the four plots were 6.43, 4.09, 5.99 and 3.01%; rBias were −0.71, 0.38, 2.05 and −0.61%; while RMSE were 1.23, 0.78, 1.01 and 0.88 cm, respectively. The main factors that affect the accuracy of DBH measurement is that the DBH of the measured trees is not in a regular circle at the DBH, while DBH measured by means of photogrammetry is in the diameter of the cross section, which may induce some errors when the DBH is measured using the DBH caliper. In this study, the images of trees

obtained through the tree-measuring instrument can be compared in real time, while the non-contact measurement method saves working hours and labor force. In addition, the non-contact method is especially useful for trees on steep slopes that cannot be reached. Earlier studies [18,22,24,25,39] have investigated methods for obtaining photos by ground photogrammetry using a separate digital camera, which requires post-processing and cannot be performed in real time. Compared with the method of extracting DBH from point-cloud data [16–19,30,31], this method can quickly extract DBH values in real time, with very little data. This method also avoids the processing of data on a computer, which not only reduces the cost of data analysis but also improves the work efficiency. Few studies to date have utilized land-based photogrammetry for extracting stand factors to assist forest resource inventory and improve work efficiency [17,23,44].

The tree height measurement method used in this paper is similar to the traditional tree height measurement method, which calculates the tree height by calculating the position of the stem and the distance between the tree-measuring instrument as well as the angle at the bottom and the top of the tree. The results show that the deviation was about −0.09–0.22 m, and the RMSE value was about 0.32–0.58 m. The results also show that when the tree height was in the range of 5–20 m, the measurement result had a high accuracy. The measurement error of tree height was mainly caused by the accuracy of the altitude sensor (the accuracy of the angle sensor is 1 degree) and the effect of canopy occlusion on the measurement of tree height. Earlier studies [22,27] have focused on the use of mobile phone angle sensors to obtain tree heights, but there is a large deviation, and the accuracy of data cannot be guaranteed. The high accuracy of the equipment in this paper is due to the integration of a high-precision angle sensor and the calculation of horizontal distances by means of two-point positioning. Compared with the height measuring equipment of the laser rangefinder, it can effectively avoid the inaccuracy of ranging and aiming caused by the laser being blocked. A laser relascope, as a well-known conventional instrument, was evaluated based on the literature, which reported a deviation of −0.016 m and a standard deviation of 0.19 m.

In terms of the working efficiency and cost, the traditional measurement methods require the operator to carry more equipment that is relatively bulky when surveying a forest, while the equipment designed in this paper can realize the integration of internal and field survey functionalities. The time required to complete the measurement of a sample site with the instrument designed in this study was about 60 min, while the traditional measurement method requires at least 120 min or more. The time required is therefore only half that of traditional survey methods, reducing the working time of the staff both in the office and in the field. Due to the high integration of the equipment, the cost of the instrument is not only one-fifth that of traditional measuring equipment, but the number of personnel required is also only one-third of traditional survey methods.

Although this instrument is a valuable new option for plot surveys, it still has certain limitations. First, in high-density forest areas, the instrument's communication signal is disturbed, and the positioning accuracy of the instrument is affected. Secondly, due to the complexity and diversity of forest plot conditions, some inevitable factors caused by unobservable conditions may still require traditional instruments to assist in the measurement. In order to accurately locate trees under the forest canopy, the four anchors of the instrument should be arranged as evenly as possible around the plot, which can reduce the positioning error of the instrument.

## 6. Conclusions

In this paper, we used a UWB positioning module as a basis to develop and test a forest plot measurement instrument. We proposed a combination of ground photogrammetry sensor technology and wireless communication technology as a new method for forest resource inventory. The test results show that using the method developed in this paper can effectively overcome the time-consuming and laborious procedures of traditional approaches using integrated equipment. When surveying a sample plot, it is not necessary

to use tools such as a forest compass and tape measure. When measuring trees, there is no need to use a hypsometer, DBH caliper or other external independent measuring tools. The ground-based measuring tools can quickly measure the stumps, saving the time and cost of the survey. The results show that the estimates of the stem location, DBH and tree height are accurate.

In further research, the instrument should be tested under complex forest conditions of different stand densities, different shrub coverage and different forest ages. Future research should also focus on integrating the functions of other forest attributes, such as stand storage and stand density.

**Author Contributions:** Data curation, Z.Z.; Formal analysis, Y.W.; Funding acquisition, Z.F. and Z.Z.; Methodology, Z.Z. and J.L.; Project administration, Z.F.; Supervision, J.L.; Writing original draft, Z.Z. All authors have read and agreed to the published version of the manuscript.

**Funding:** This research was funded by the Research Fund of Tangshan Normal University (No. 2022B05) and the National Natural Science Foundation of China (Grant number U1710123).

**Institutional Review Board Statement:** Not applicable.

**Informed Consent Statement:** Not applicable.

**Data Availability Statement:** Not applicable.

**Acknowledgments:** We are grateful to the staff of the Precision Forestry Key Laboratory of Beijing, Beijing Forestry University. We are also very thankful for the prompt response of the journal of Sustainability and its reviewers. The reviewer comments were highly impactful for improving the manuscript draft.

**Conflicts of Interest:** The authors declare no conflict of interest.

## Appendix A

The highly integrated tree-measuring instrument consists of a CPU, RAM, ROM, a UWB module, a laser range finder, a GPU, a touch-control screen, a charge-coupled device (CCD) camera, an antenna, a three-axis gyroscope, a three-axis accelerometer, a three-axis electronic compass, a gyroscope, a Bluetooth chip, a Wi-Fi chip and a power source, which are housed in an aluminum alloy case (Figure A1).

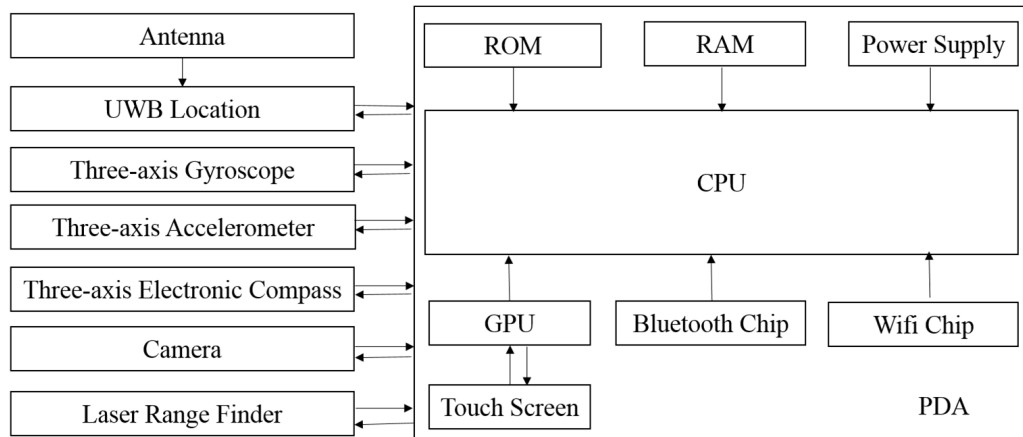

**Figure A1.** Schematic of the hardware structure design of the Tree-measuring Instrument.

## Appendix B

**Table A1.** Details of the parameters of the dendrometer.

| Parameters | Weight (g) | Size (mm) | Price (dollar) | Index | Range | Time of Assembly | Working Temperature |
|---|---|---|---|---|---|---|---|
| UWB location | 100 * 5 | 30 * 62 * 10 | 40 * 5 | ±5 cm | 450 m | | |
| tripod | 800 * 5 | 600 * 150 * 150 | 10 * 5 | - | - | | |
| Aluminum alloy shell | 265 * 1 | 170 * 100 * 95 | 30 * 1 | - | - | | |
| Attitude sensor | Integration in UWB | - | - | 1° | - | 15 min | −20~85 °C |
| Laser range finder | 120 * 1 | 100 * 40 * 30 | 20 * 1 | ±2 cm | 100 m | | |
| PDA | 150 * 1 | 150 * 80 * 9 | 100 * 1 | - | - | | |
| CCD camera | 186 * 1 | 68 * 68 * 60 | 100 * 1 | 2000 pixel | - | | |
| battery | 380 * 5 | 100 * 80 * 12 | 10 * 5 | 8000 mA | - | | |
| total | 7111 g | 600 * 300 * 180 bag | 550 | - | - | - | |

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
