# Peer review of "Development and Testing of a New UWB Positioning Measurement Tool to Assist in Forest Surveys"

_sustainability, doi:10.3390/su142417042_

Round 1
Reviewer 1 Report
This paper examines use of a multifunctional, high-precision, real-time positioning intelligent tree-measuring instrument that integrates plot set-up, DBH measurement, tree height measurement, and tree position measurement. The introduction is comprehensive and well written. The methods and results are also well described in the manuscript and the discussion and conclusions are supported by the results. Overall, the paper is good, the manuscript can be considered a valuable contribution to the literature and should be considered for publishing.Author Response
Please see the attachment.

Reviewer 2 Report
L 46: "... in areas of interest..." term should be changed as "... in stands of interest...".
L 50: "... forest surveys..." term should be changed as "... forest inventory...".
L 54: "... forest resource surveys..." term should be changed as "... forest resource inventory...".
L 89: Who are the other scholars? They should be cited.
L 114: "2.1.1. Subsubsection" term should be removed.
L 141: "... the above equation (2)." term should be changed as "... the equation (2).".
L 156: The sentence "The additional anchors UWB positioning device battery (see figure 2b)." should be checked. It has no verb.
L 205: Equation B?
L 212: The current name and synonim are the same (Cotinus coggygria).
L 214: "... four 25.82 × 25.82 m square plots..." term should be changed as "... four 25.82 × 25.82 m square (667 m2) plots...".
L 224: Is there any threshold for plot size to call "large"? It should be explained.
L 243: How will the position error be determined to meet the accuracy requirements?
L 269: "... using equation (6) and..." term should be changed as "... using equation (4) and...".
Equation 5: "D" should be changed as "DBH".
L 308: The distance from the tree should be clearly stated.
Equation 6: It should be moved L 314.
L 349: "0.14 m" should be changed as "0.16 m".
L 350: "0.09-0.17 m" should be changed as "0.07-0.16 m".
L 350: "0.07-0.17 m" should be changed as "0.09-0.12 m".
L 364: "... correct values." term should be changed as "... reference values.".
L 368-369: What are these requirements? They should be cited.
L 373: Figures 13a-d?
L 391: Figures 14a-d?
L 437: "... principle." term should be changed as "... principle, respectively.".
L 454-455: The RMSE values should be checked.
Reviewer 3 Report
Review report
The article entitled Development and Testing of a New UWB Positioning Measurement Tool to Assist in Forest Surveys by Ziyu Zhao et al. refers to the methodology of conducting basic dendrometric measurements (dbh, height and location of trees) within forest sample plots. It contains a detailed description of the UWB technology concept and the instruments used for these measurements, as well as the measurement procedures themselves. The article also includes a comparison of the results of measurements made with the proposed methodology with the results of measurements of the same parameters made with the NTS-391 total station to verify the correctness and accuracy of the UWB technology and the proposed hardware configuration. This comparison showed a high agreement of the measurement results of both methodologies, which proves the possibility of using the UWB technology to measure the basic dendrometric parameters of forest trees.
The structure of the article is correct, logical and coherent. The description of the methodology is exhaustive and clear. The presentation of the results is clear and illustrated with appropriate diagrams. The discussion is factual and the conclusions are correct and confirmed by the research results. All this speaks for the high rating of the manuscript submitted for review. However, the manuscript's thematic scope and nature (it is essentially a technical description of the method and its application) is far from the topic of interest to the journal Sustaiability (see Sustainability Aims & Scope). I therefore recommend submitting the manuscript to another journal whose area of interest is better suited to the theme of the article. That is why I refused to give an "overall merit" and finally marked the "rejection" as a general recommendation. Of course, the final decision in this matter should be made by the editor of the journal and not by me.
Round 2
Reviewer 3 Report
Review Report 2
As I wrote in an earlier review, I have no complaints about the quality of the article. The explanations given by the author and the Academic Editor have convinced me to withdraw my earlier recommendation to reject the article and publish it in another journal.
I would just like to take this opportunity to draw the authors' attention to typos and missing spaces between words in the text. Below are some examples:
Line 79: no space before the parenthesis
Line 88: no space before parenthesis with added reference
Line 211: no space
Line 238: Measuremen.... It should be "Measurement" as in line 237. A similar error occurs in lines 280, 289, where there is also no space after the subsection numbers.
Line 312: remove the dot and format the font size
Line 337: double comma
Line 353: no space before the parenthesis
Table 3: no space before the bracket
Line 377: no space
Also, in subsection 3.2.2, I would suggest replacing the numbering of the next level of subsections with underlining of the titles. Number 1 already has the chapter Introduction, number 2 Theory and Technology, and number 3 the chapter Materials and Methods
